# Metabolomic Analysis of the Response of *Haloxylon ammodendron* and *Haloxylon persicum* to Drought

**DOI:** 10.3390/ijms24109099

**Published:** 2023-05-22

**Authors:** Fang Yang, Guanghui Lv

**Affiliations:** 1School of Ecology and Environment, Xinjiang University, Urumqi 830017, China; yf1121@xju.edu.cn; 2Key Laboratory of Oasis Ecology, Ministry of Education, Urumqi 830017, China; 3Xinjiang Jinghe Observation and Research Station of Temperate Desert Ecosystem, Ministry of Education, Jinghe 833300, China

**Keywords:** drought stress, *Haloxylon ammodendron*, *Haloxylon persicum*, metabolomics

## Abstract

*Haloxylon ammodendron* and *Haloxylon persicum*, as typical desert plants in arid areas, show strong drought tolerance and environmental adaptability and are therefore ideal model plants for studying the molecular mechanisms of drought tolerance. A metabolomic analysis of *H. ammodendron* and *H. persicum* in their natural environment is lacking, and their metabolic response to drought therefore remains unclear. To elucidate the response of *H. ammodendron* and *H. persicum* to drought at the metabolic level, a non-targeted metabolomics analysis was carried out herein. Under a dry environment, *H. ammodendron* exhibited 296 and 252 differentially expressed metabolites (DEMs) in the positive and negative ion modes, respectively, whereas 452 and 354 DEMs were identified in the positive and negative ion modes in *H. persicum*, respectively. The results indicated that *H. ammodendron* responds to drought by increasing the content of organic nitrogen compounds and lignans, neolignans, and related compounds, and reducing the content of alkaloids and derivatives. By contrast, *H. persicum* adapts to the dry environment by increasing the content of organic acids and their derivatives and reducing the content of lignans, neolignans, and related compounds. In addition, *H. ammodendron* and *H. persicum* improved their osmoregulation ability, reactive oxygen species detoxification ability, and cell membrane stability by regulating the key metabolic pathways and anabolism of associated metabolites. This is the first metabolomics report on the response of *H. ammodendron* and *H. persicum* to drought in their natural environment, providing a foundation for the further study of their regulatory mechanisms under drought stress.

## 1. Introduction

The tolerance and sensitivity of plants to stress is a complex biological process, the most important feature of which is the change in metabolism. Plants can induce a variety of specialized metabolites by regulating metabolic networks to defend against drought stress [1]. Under drought stress, the synthesis of the genes of related metabolites in plants is induced, promoting the accumulation of related metabolites, mainly including primary metabolites (carbohydrates, lipids, amino acids, nucleic acids, and organic acids) and secondary metabolites (such as alkaloids, phenols, quinones, xanthones, and terpenes) [2,3]. These substances play an important role in protecting plants from environmental stress and maintaining cell stability [4,5]. There are many types of specialized metabolites in plants, and, therefore, systematically studying and identifying the changes in metabolites under drought stress is a complex endeavor.

Understanding physiology at the whole-cell level is a difficult task. However, “omics” technology has provided an unprecedented way to understand the underlying structure and function of biological networks [6,7]. Various levels of cell structure have been revealed through genomics, transcriptomics, and proteomics, but another layer of information is needed to clarify the relationship between the genotype and phenotype [8]. Metabolomics is expected to fill this gap and provide quantitative information about the level of intracellular metabolites, which represent the highest level of functional components in the cell process [8,9]. The systematic identification and quantification of small and low-molecular-weight compounds in cells is the goal of metabolomics [10,11].

In recent years, metabolomics has been gradually used to study the metabolic regulation network mechanisms of plant responses to drought stress. Wei et al. (2020) found that three metabolites (galactoside, neoxanthin, and arbutin) in safflower are related to drought tolerance, and the regulation mechanism varies with genotype [12]. Zhao et al. (2021) analyzed *Cichorium intybus* and found that its drought tolerance mechanism involves glycolysis, phenol metabolism, tricarboxylic acid cycle, glutamic-acid-mediated proline synthesis, amino acid metabolism, and unsaturated fatty acid synthesis [13]. Metabolites can also accumulate under drought stress. Researchers have identified the key metabolic pathways and metabolites of *Glycyrrhiza uralensis* [14], rice [15], and soybean [16] in response to drought stress and have clarified their associated metabolic mechanisms.

*H. ammodendron* and *H. persicum* belong to the genus *Haloxylon* of Chenopodiaceae, with developed main and lateral roots and degenerated leaves, and photosynthesis is carried out by assimilating branches [17], and they are extremely drought-tolerant species. *H. ammodendron* and *H. persicum* can reduce wind speed, improve forest microclimate, and promote the settlement and growth of other desert plants [18]. Therefore, they play an important role in maintaining the structure and function of the entire ecosystem where they grow [19]. Although *H. ammodendron* and *H. persicum* belong to the same genus, their distribution in arid areas is quite different. *H. ammodendron* grows in fixed sand dunes, and hardly in flowing and semi-flowing sand dunes, while *H. persicum* is the opposite. These trees are rapidly approaching extinction due to a number of factors [19,20] and the molecular mechanisms underlying the long-term responses of *H. ammodendron* and *H. persicum* to drought stress at the metabolic level in the natural environment remain unclear. To explore the changes of metabolites in *H. ammodendron* and *H. persicum* under drought stress, the responses of *H. ammodendron* (humid (HS): soil water content 9.70–15.00%; drought (LS): soil water content 2.41–4.00%) and *H. persicum* (humid (HB): soil water content 3.38–5.12%; drought (LB): soil water content 1.05–3.11%) in humid and arid ecological environments were examined. This study used liquid chromatography-tandem mass spectrometry (LC-MS/MS) to compare and analyze the changes in the metabolites of *H. ammodendron* and *H. persicum* under drought stress, screen key metabolites and key metabolic pathways, and reveal the drought-tolerant metabolic mechanisms of these two species.

## 2. Results

### 2.1. Metabolite Characterization

Metabolite detection was divided into positive ion and negative ion modes. In the positive ion mode, *H. ammodendron* and *H. persicum* had 897 and 932 DEMs, respectively, which were divided into 11 types, while in the negative ion mode, 642 and 669 DEMs were identified, respectively, which were divided into nine types (Figure 1). In both ionization modes, benzenoids (3.79% and 3.86%; 3.27% and 3.74%), lipids and lipid-like molecules (7.13% and 8.48%; 7.63% and 8.67%), organic acids and derivatives (4.79% and 3.86%; 4.83% and 5.38%), organic oxygen compounds (2.56% and 2.25%; 3.58% and 4.19%), organic heterocyclic compounds (5.35% and 5.26%; 2.96% and 4.63%), and phenylpropanoids and polyketides (8.14% and 8.05%; 5.61% and 6.88%) were the main metabolites.

### 2.2. Quantitative Analysis of Metabolites and Sample Quality Control Analysis

The mass spectrum peaks of all substances in *H. ammodendron* and *H. persicum* in arid and humid soil moisture environments were integrated, and the mass spectrum peaks of the same metabolite in different samples were integrated and corrected to ensure quantification accuracy. By scanning and overlapping the total ion flow chromatograms (TIC) of the mass spectrometric analysis of different quality control (QC) samples of *H. ammodendron* and *H. persicum* in the positive and negative ion modes (Figure 2 and Figure 3), it was found that the repeatability of the metabolite extraction and detection in this study was high, indicating that the experimental instrument and data collection were stable. There were obvious chromatographic differences between the arid and humid soil moisture environments, and the experimental results were deemed reliable.

We performed a PCA analysis on the peaks extracted from all experimental and QC samples. The greater the difference in the QC samples, the better the stability of the entire method. The higher the data quality, which is reflected in the PCA analysis chart, the more the distribution of QC samples will gather together. The peaks extracted from all experimental samples and QC samples of *H. ammodendron* and *H. persicum* were subjected to PCA (Figure 4). Under the positive and negative ion modes, the total cumulative contribution rates of the first two principal components (PCs) in the PCA analysis of *H. ammodendron* were 67.8% and 67.85%, respectively (Figure 4A,B), while, in *H. persicum,* these values were 82.96% and 79.12%, respectively (Figure 4C,D). This indicated that drought had a significant impact on the changes in metabolites in the two species under wet and dry environments. Furthermore, the QC samples grouped together, indicating that the method was reliable and the data quality was high, and therefore the data could be used for subsequent analysis.

We calculated Pearson’s correlation coefficients between QC samples based on relative quantitative values of metabolites. The Pearson’s correlation coefficient between the QC samples was calculated based on the relative quantitative value of the metabolites (Figure 5). In the positive and negative ion modes, the Pearson’s correlation coefficient R^2^ values between the QC samples of *H. ammodendron* and *H. persicum* reached more than 98%, indicating that the instrument exhibited good stability, high reliability, and high metabolite data quality during the entire detection process.

### 2.3. Metabolite Pathway and Classification Annotation

#### 2.3.1. KEGG Pathway Annotation

Metabolic pathway analysis showed that the 897 and 642 metabolites identified in *H. ammodendron* under positive and negative ion modes, respectively, were mainly involved in three major categories, namely, environmental information processing, genetic information processing, and metabolism (Figure 6A,B). Among them, the metabolites involved in metabolism were most abundant. In addition, in the positive ion mode, 278 metabolites of *H. ammodendron* were annotated into 50 pathways, the largest category being “metabolic pathways (map01100)”, followed by “Biosynthesis of secondary metabolites (map01110)”, “Flavonoid biosynthesis (map00941)”, “Phenylpropanoid biosynthesis (map00940)”, “ABC transporters (map02010)”, “Phenylalanine metabolism (map00360)”, “Purine metabolism (map00230)”, and “Tropane, piperidine, and pyridine alkaloid biosynthesis (map00960)”. Under the negative ion mode, 190 metabolites of *H. ammodendron* were annotated into 47 pathways. The largest category was “Metabolic pathways (map01100)”, followed by “Biosynthesis of secondary metabolites (map01110)”, “Purine metabolism (map00230)”, “Pyrimidine metabolism (map00240)”, “Flavonoid biosynthesis (map00941)”, “Glyoxylate and dicarboxylate metabolism (map00630)”, “Phenylpropanoid biosynthesis (map00940)”, and “Flavone and flavonol biosynthesis (map00944)”.

As with *H. ammodendron*, the 932 metabolites of *H. persicum* in the positive ion mode were also divided into three categories (Figure 6C), while 669 metabolites in the negative ion mode were annotated to four major categories of pathways; namely, cellular processes, environmental information processing, genetic information processing, and metabolism (Figure 6D). Similar to *H. ammodendron*, most metabolites were involved in metabolism, and the cell process was mainly transport and catabolism. In addition, 288 metabolites were annotated into 47 pathways in the positive ion mode, the largest category being “Metabolic pathways (map01100)”, followed by “Biosynthesis of secondary metabolites (map01110)”, “Flavonoid biosynthesis (map00941)”, “Phenylpropanoid biosynthesis (map00940)”, “Stilbenoid, diarylheptanoid and gingerol biosynthesis (map00945)”, “Flavone and flavonol biosynthesis (map 00944)”, “Isoquinoline alkaloid biosynthesis (map00950)”, and “Phenylalanine metabolism (map 00360)”. In the negative ion mode, 230 metabolites were annotated into 50 pathways, the largest category being “Metabolic pathways (map01100)”, followed by “Biosynthesis of secondary metabolites (map01110)”, “Phenylpropanoid biosynthesis (map00940)”, “Glyoxylate and dicarboxylate metabolism (map00630)”, “Phenylalanine metabolism (map00360)”, and “Purine metabolism (map00230)”.

#### 2.3.2. Multivariate Statistical Analysis

Under the positive and negative ion modes, the total cumulative contribution rates of the first two PCs were 72.49% and 65.62%, respectively, for *H. ammodendron* (Figure 7A,B), and 72.04% and 73.54%, respectively, for *H. persicum* (Figure 7C,D), indicating that there were significant differences in the metabolite changes in *H. ammodendron* and *H. persicum* under dry and humid soil environments.

Similar to PCA, PLS-DA can maximize the differentiation between groups and help to identify differential metabolites. Under the positive and negative ion modes, the R2Y and Q2 values of the two comparison groups of LS vs. HS and LB vs. HB were close to 1 and had an R2Y > Q2 (Figure 8A,C,E,G), indicating that the PLS-DA model was stable and reliable.

The grouping of each sample was randomly disordered before modeling and prediction. Each model corresponds to a set of R2 and Q2 values. Their regression lines were obtained from the Q2 and R2 values after 200 times of disordering and modeling. Under the positive and negative ion modes, the R2 values of the *H. ammodendron* comparison group LS vs. HS and *H. persicum* comparison group LB vs. HB (Figure 8B,D,F,H) were greater than the Q2 data, and the intercept between the Q2 regression line and *Y*-axis was less than 0. This indicated that the model was not overfitted and that it could adequately describe the samples of the two species under the two conditions, and that the data could be used for further analysis.

#### 2.3.3. Identification of DEMs

In the positive and negative ion modes, 296 (183 upregulated, 113 downregulated) and 252 (119 upregulated, 133 downregulated) DEMs (Figure 9A, Appendix A) were identified in *H. ammodendron*, while 452 (254 upregulated, 198 downregulated) and 354 (173 upregulated, 181 downregulated) DEMs were identified in *H. persicum* (Figure 9C, Appendix A). The DEMs identified in *H. ammodendron* in the positive ion mode were classified into 11 categories (Figure 9B), in which the upregulated DEMs were mainly organic nitrogen compounds and the downregulated DEMs were mainly alkaloids and derivatives. In the negative ion mode, the DEMs were classified into 10 categories, among which lignans, neolignans, and related compounds were upregulated, and alkaloids and derivatives were downregulated.

The DEMs identified in *H. persicum* in the positive ion mode were classified into 12 categories (Figure 9D), in which the upregulated DEMs included organic compounds and the downregulated ones included lignans, neolignans, and related compounds. The DEMs identified in the negative ion mode were classified into nine categories, among which organic acids and derivatives were upregulated.

### 2.4. KEGG Enrichment Analysis of the DEMs

In the positive ion mode, the main metabolic pathways of the DEMs between LS and HS included “Flavonoid biosynthesis”, “Isoquinoline alkaloid biosynthesis”, “Tyrosine metabolism”, and “Phenylalanine metabolism” (Figure 10A). Among these, the most significantly enriched metabolic pathway was “Flavonoid biosynthesis”, and the main DEMs were hesperidin (upregulated), dihydrokaempferol (upregulated), galangin (upregulated), pinocembrin (upregulated), neohesperidin (downregulated), dihydroquercetin (upregulated), kaempferol (downregulated), and vitexin (upregulated). In the negative ion mode, “Biosynthesis of unsaturated fatty acids”, “Isoquinoline alkaloid biosynthesis”, and “Glycolysis/Gluconeogenesis” were primarily enriched (Figure 10B). Among these, the most significantly enriched metabolic pathway was “Biosynthesis of unsaturated fatty acids “, and the DEMs involved included lignoceric acid (downregulated), erucic acid (downregulated), docosanoic acid (downregulated), and arachidic acid (downregulated).

In the positive ion mode, the main metabolic pathways of the DEMs between LB and HB were “Tropane, piperidine, and pyridine alkaloid biosynthesis”, “Isoquinoline alkaloid biosynthesis”, and “Pyrimidine metabolism” (Figure 10C). Among them, the most significantly enriched metabolic pathway was “Tropane, piperidine, and pyridine alkaloid biosynthesis”. The enriched DEMs included pipecolic acid (downregulated), lupine (downregulated), senecinine (upregulated), tropine (downregulated), and L-phenylalanine (upregulated). The metabolic pathways that were mainly enriched in the negative ion mode were “Biosynthesis of secondary metabolites”, “Ascorbate and aldoate metabolism”, and “Glutathione metabolism” (Figure 10D). Among these, the most significantly enriched metabolic pathway was “Biosynthesis of secondary metabolites”, and the main enriched DEMs were luteolin (downregulation), caffeine (upregulated), coniferol (upregulated), D-ribose 1,5-bisphosphonate (upregulated), and gluconolactone (upregulated).

Under the positive ion mode, L-dopa and salicylic acid, which were the most differentially expressed metabolites in *H. ammodendron*, participated in three and two (phenylalanine metabolism and plant hormone signal transduction) metabolic pathways, respectively. In the negative ion mode, β-D-fructose 6-phosphate, D-mannose 6-phosphate, abscisic acid, and L-tryptophan participated in five, three, three, and four metabolic pathways, respectively (Table 1).

In the positive ion mode, α-ketoglutaric acid, L-saccharophane, and L-phenylalanine in *H. persicum* participated in six, four, and four metabolic pathways, respectively. In the negative ion mode, L-glutamic acid, succinic acid, and citric acid participated in eight, seven, and six metabolic pathways, respectively (Table 1).

### 2.5. Construction of the Metabolic Pathways of H. ammodendron and H. persicum in Response to Drought Stress Using Metabolomics

To clarify the metabolic response of *H. ammodendron* and *H. persicum* to drought, the key metabolic pathways of drought tolerance were constructed (Figure 11 and Figure 12). For *H. ammodendron*, drought significantly increased the content of substances in glycometabolism and amino acid metabolism. Among them, β-D-fructose 6-phosphate, D-mannose 6-phosphate, L-tryptophan, and S-sulfo-L-cysteine increased significantly, indicating that *H. ammodendron* accumulated some sugars and amino acids and their derivatives to maintain cell turgor and provide raw materials for the synthesis of new proteins in the dry environment. At the same time, these enhanced the abscisic acid (ABA) signal transduction pathway of *H. ammodendron*, but inhibited the salicylic acid signal pathway, indicating that the ABA signal transduction pathway plays an important role in the response of *H. ammodendron* to drought. In addition, the content of the secondary metabolites phenylglyoxylic acid, benzoic acid, pseudoephedrine, and rosmarinic acid in *H. ammodendron* increased significantly, while the content of L-dopa, tyramine, and 3,4-dihydroxybenzaldehyde decreased significantly. These secondary metabolites can be used as both important osmoregulatory substances and reactive oxygen species (ROS) scavengers. They probably play an important role in improving the osmoregulation ability and ROS detoxification ability of *H. ammodendron*.

For *H. persicum*, drought significantly increased the content of substances in carbohydrate metabolism, amino acid metabolism, secondary metabolite synthesis, and transcriptional and post-transcriptional regulation. Among them, citric acid, succinic acid, α-alpha-ketoglutaric acid, L-saccharopine, L-phenylalanine, L-glutamic acid, L-histidine, shikimic acid, glyoxylate, pantethine, 2-oxoglutarate, adenine, and senecionine increased significantly, while the content of the secondary metabolite lupinine was significantly reduced. It is clear that *H. ammodendron* and *H. persicum* can enhance their drought tolerance by continuously adjusting the content of key metabolites in carbohydrate metabolism, amino metabolism, and secondary metabolism.

## 3. Discussion

To survive in harsh environments, plants have evolved a complex system acting at multiple levels to sense external signals and timeously transmit stress signals, leading to a series of reactions at the morphological, physiological, biochemical, and molecular levels [21]. Metabolomics analysis can explain the regulatory signals related to transcription, translation, and post-translation processes [22,23]. Generally, key metabolites are metabolites that are related to potential biochemical pathways, enzymes, or genes [24]. In this study, under the positive and negative ion modes, 296 and 252 DEMs were, respectively, identified in *H. ammodendron*, 28 of which were identified as key candidate metabolites (Appendix A). In *H. persicum*, 50 DEMs were identified as key candidate metabolites. In general, the contents of organic nitrogen compounds, lignans, neolignans, and related compounds in *H. ammodendron* increased, while the contents of alkaloids and their derivatives decreased. The content of organic compounds, organic acids, and their derivatives in *H. persicum* increased, while the content of lignans, neolignans, and related compounds decreased. Research shows that metabolites in different drought-tolerant species have both similar and different change rules, which will affect their drought tolerance [1]. In this study, the metabolic pathways significantly enriched in *H. ammodendron* were “Flavonoid biosynthesis” and “Unsaturated fatty acid biosynthesis”, while in *H. persicum* these were “Tropane, piperidine, and pyridine alkaloid biosynthesis” and “Secondary metabolite biosynthesis”. The changes in metabolites in *H. ammodendron* and *H. persicum* in response to drought were analyzed to clarify their drought-tolerance mechanism and explore the key metabolites of drought tolerance. The identified metabolites were functionally classified to reveal the drought-tolerant metabolic spectrum of *H. ammodendron* and *H. persicum*.

### 3.1. Amino Acids and Their Derivatives

In most cases, increases in most amino acids can be attributed to decreased protein synthesis or enhanced protein decomposition induced by drought [25]. Good et al. (1994) found that, during drought stress in *Brassica napus*, the content of most amino acids in the leaves increased linearly, being 5.9 times higher than the control on average [26]. In this study, drought induced the production of many amino acids and their derivatives in *H. ammodendron* and *H. persicum*, and their upregulation in both species was high, indicating that these amino acids and their derivatives play an important role in maintaining the intracellular osmoregulation and integrity of the protein structure [27], which is an important drought response mechanism of *H. ammodendron* and *H. persicum*. Some studies have found that the content of free amino acids in plants increases significantly under drought stress [28]. Under a dry environment, only L-tryptophan increased significantly in *H. ammodendron*, while L-glutamic acid, L-tryptophan, L-histidine, L-phenylalanine, proline, (S)-glutamic acid, and L-aspartic acid increased significantly in *H. persicum*. This indicated that *H. persicum* has a strong ability to activate the effective molecules and biochemical processes to maintain the osmotic regulation required for plant growth. In addition, L-tryptophan and L-histidine in *H. ammodendron* and L-phenylalanine, L-glutamic acid, L-histidine, and L-tryptophan in *H. persicum* participate in the biosynthesis of aminoacyl tRNA, indicating that drought stress promotes the production of metabolites in the translational process of the two species. Michaletti et al. (2018) found that all aromatic amino acids (tryptophan, phenylalanine, and tyrosine) in spring wheat accumulated under drought stress. Aromatic amino acids are synthesized in plants through the shikimic acid pathway. Aromatic amino acids are oxidation targets, and their free form may have protective effects against ROS [29]. The increase in the contents of tryptophan and phenylalanine in the two species in this study also showed that the antioxidant mechanism could be mobilized to eliminate ROS in response to drought stress.

Under drought stress, there were significant differences in the glutathione metabolism pathway between drought-tolerant *Selaginella lepidophylla* and drought-sensitive *Selaginella moellendorfii* [30]. The relative content of several intermediate products in drought-sensitive *Selaginella vulgaris* was higher, such as glycine, cysteine, 5-oxoproline, and glutamic acid. The drought-resistant *S. vulgaris* accumulated more oxidized glutathione and c-glutamyl amino acid [30]. In this study, glutathione and glutathione metabolic intermediates in the two species increased significantly under drought stress, but their metabolic intermediates changed differently due to species differences, indicating that the antioxidant stress capacity of the two species differs.

### 3.2. Lipid Metabolites

Increasing evidence confirms that plant species that are more tolerant to drought stress maintain higher levels of unsaturated fatty acids, such as hexadecenoic acid, palmitic acid, stearic acid, and linolenic acid, all of which will interfere with membrane fluidity and cell function damage [31,32]. One study showed that, under the two water conditions, the content of one monounsaturated fatty acid and four polyunsaturated fatty acids in drought-tolerant *S. vulgaris* increased significantly [30]. These changes may reflect the ability to increase membrane fluidity by changing the level of unsaturated fatty acids [33,34]. For example, the overexpression of two fatty acid desaturases in tobacco increases the tolerance to drought and osmotic stress [35]. In the present study, 13-hydroperoxylinoleic acid (13-HPODE) was significantly increased in *H. ammodendron*, and docosahexaenoic acid was slightly increased in *H. persicum*, indicating that drought stress affected the drought tolerance of the two species by influencing the content of unsaturated fatty acids.

Membrane integrity is considered to play an important role in drought tolerance [36,37]. Under drought stress, the content of phospholipids in *Morus alba* L. leaves decreased [22]. The content of 1-(sn-glycero-3-phospho)-1D myoinositol in soybean leaves significantly decreased under drought stress [38]. In this study, six types of glycerol phospholipid metabolites (LPG 18:3, LPG 18:2, LPG 18:4, LPA 16:0, LPG 16:0, and LPA 16:1) increased significantly in *H. persicum*, indicating the changes in the biological membrane components of *H. ammodendron* under drought stress.

### 3.3. Secondary Metabolites

Under drought stress, the contents of organic acids, benzoic acid derivatives, hydroxycinnamic acid derivatives, isoflavones, and flavonoids in *H. ammodendron* and *H. persicum* changed significantly. Among them, organic acid metabolites play an important role in improving plant osmoregulation, and one and three DEMs were identified as organic acid substances in *H. ammodendron* and *H. persicum,* respectively. The contents of α-ketoglutaric acid, citric acid, and succinic acid were significantly increased in *H. persicum*, indicating that the above metabolites played an important role in the response of *H. persicum* to drought stress. Moreover, these three substances are the main intermediate metabolites in the TCA cycle. The increase in their content shows that *H. ammodendron* needs sufficient energy to adapt to drought, and the TCA cycle plays an important role in this process.

Studies have shown that antioxidants can act as oxygen scavengers to interfere with the oxidation process caused by various stresses, and so the tolerance to drought stress may be related to the improvement of antioxidant potential [39,40]. In many plants, phenylpropanoid metabolic pathways are known to respond to stress and reduce UV damage and oxidative stress [41]. In this study, there were six DEMs involved in the synthesis and metabolism of phenylpropanes in *H. ammodendron* and *H. persicum*. Among them, the contents of chrysin, galangin, and 3,5,7-trihydroxyflavone in *H. ammodendron* increased significantly, while the contents of corilagin, luteolin, and kaempferol decreased significantly. The contents of isorhamnetin and galangin in *H. persicum* increased significantly, while the contents of ellagic acid, quercetin, luteolin, and kaempferol decreased significantly. This suggested that the phenylpropanoid synthesis pathway plays a key role in the response of *H. ammodendron* and *H. persicum* to drought stress. Some studies have shown that changes in phenolic compounds are positively related to their anti-free radical activity and reduction ability [42,43,44]. Osmotic stress reduces the contents of total phenols and phenolic acids in plant seeds and also reduces their antioxidant capacity [45]. In this study, catechol in *H. ammodendron* was upregulated under drought stress, while coniferol and 4-ethylphenol were downregulated. In *H. persicum,* eugenol, catechol, coniferyl alcohol, vanillin alcohol, and homovanillic acid were upregulated, indicating that these substances may be the key substances affecting the antioxidant capacity of *H. ammodendron* and *H. persicum*.

Flavonoid metabolites are the main defense substances of plants. As the primary components of secondary metabolism, they play a vital role in resisting environmental stress [46]. Studies have shown that flavonoid metabolites can alleviate the damage caused by drought stress [45]. The abundance of flavonoids derived from amino acids (such as apigenin, luteolin, and naringin) in drought-tolerant *S. lepidophylla* increased under two water conditions [30]. Drought induced an increase in the total flavone content in tea and decreased the content of polyphenols, catechins, caffeine, theanine, and some free amino acids [47]. However, the present study found that drought stress inhibited the production of flavonoids (such as myricetin 3-O-galactoside and sage phenol-7-O-glucoside) in *H. persicum*, which was contrary to that in *H. ammodendron*, indicating that drought stress had different effects on secondary metabolism in these two species. On the other hand, 4′,6,7-trihydroxyisoflavone is mainly involved in isoflavone biosynthesis and flavonoid biosynthesis and metabolism [45]. The content of this substance was significantly reduced in *H. persicum*, indicating that drought stress significantly inhibited the biosynthesis and metabolism of isoflavone and flavonoid in *H. persicum*.

Benzoic acid is the only common component of salicylic acid derivatives that can induce stress tolerance. Benzoic acid is also more effective than salicylic acid and its derivatives at lower concentrations [48]. Exogenous benzoic acid significantly increased the net photosynthesis of soybean under drought stress by 11.54% and the content of chlorophyll a by 6.57%, thus improving its growth and yield [49]. In this study, 10 and eight DEMs were identified as benzoic acid and derivatives in *H. ammodendron* and *H. persicum*, respectively. Among them, the contents of gallic acid trimethyl ether, benzoic acid, and salicylic acid increased significantly in *H. ammodendron*, indicating that drought stress induced the accumulation of benzoic acid and salicylic acid in *H. ammodendron* to maintain normal growth.

### 3.4. Nucleosides, Nucleotides, and Analogues

Michaletti et al. (2018) analyzed the metabolome of the leaves of two spring wheat varieties under drought stress and found that drought stress had little effect on the metabolome of drought-tolerant spring wheat varieties, and purine metabolism was the basic way of change in drought-tolerant varieties [29]. Six and 13 DEMs were identified as nucleosides, nucleotides, and analogues in *H. ammodendron* and *H. persicum*, respectively. The contents of four nucleosides, nucleotides, and analogues in *H. ammodendron* increased significantly, including purine nucleotides (adenosine and inosin’-5′-monophosphate), pyrimidine nucleotides (cytidine-5′-monophosphate), and ribonucleotides (AICA ribonucleotide). The content of nine nucleosides, nucleotides, and analogues in *H. persicum* increased significantly, including flavin nucleotides, purine nucleosides, pyrimidine nucleosides, and 5′-deoxyribonucleic acid. This shows that, under drought stress, *H. ammodendron* and *H. persicum* protect nucleic acids by altering purine metabolism and pyrimidine metabolism.

### 3.5. Other Metabolites

Drought stress also affects the production of other metabolites (such as alkaloids and derivatives). Under drought stress, the contents of trigonelline and 6-acetylmorphine in *H. persicum* were significantly decreased, while the contents of berberine were significantly increased, indicating that these substances also played an important role in the response of *H. persicum* to drought stress. The contents of sanguinarine, trigonelline, and 6-acetylmorphine in *H. ammodendron* significantly decreased under drought stress, indicating that different metabolites (or the same metabolites) play different roles in response to drought stress in different species of *Haloxylon*, which may be related to the gene regulatory networks of these species. Different water use strategies employed by the two *Haloxylon* species drove *H. persicum*, but not *H. ammodendron*, to increase its photosynthetic capacity when groundwater was deeper [17]. This is also one of the reasons why there may be differences in the types of metabolites synthesized by the two species.

## 4. Materials and Methods

### 4.1. Sample Plot Setting

Plants were sampled during the vigorous plant growth season from June to July 2021 at the field scientific observation station of the Ministry of Education of the temperate desert ecosystem in Jinghe County, Xinjiang University, starting from the East Bridge Management Station of Ebinur Lake Wetland Nature Reserve. Transects (width 0.1 km × 2.0 km) were set up perpendicular to the Aqiksu River in the areas with *H. ammodendron* and *H. persicum* and labeled as transect 1 (in the distribution area of *H. ammodendron*) and transect 2 (in the distribution area of *H. persicum*). According to the research results of previous studies (by this research group) [50], two soil water environments, humid and arid, were selected. A quadrat size of 50 m × 50 m was used in each of the two soil moisture environments (*H. ammodendron*: A: humid and low salinity (HS), and B: arid and low salinity (LS); *H. persicum*: C: humid and low salinity (HB), and D: arid and low salinity (LB)) (Figure 13). We investigated the number and abundance of woody plant species in the plots and selected five individuals of *H. ammodendron* and *H. persicum* with similar individual sizes with regard to plant height, crown width, and base diameter for sampling and determination.

### 4.2. Metabolite Extraction and Identification

Metabolite extraction was performed by referring to the method of Want et al. (2010) [51]. A Vanquish UHPLC system (Thermo Fisher, Suzhou, China) and Orbitrap Q ActiveTMHF-X mass spectrometer (Thermo Fisher, Suzhou, China) were used for ultra-high-performance (UHP)LC-MS/MS analysis in Beijing Novogene Technology Co., Ltd. (Beijing, China). Samples were injected into a Hypsil Gold column (100 × 2.1 mm, 1.9 μm); a 17 min linear gradient was adopted; and the flow rate was 0.2 mL/min. The positive mode eluent was eluent A (0.1% FA in water) and eluent B (methanol). The negative mode eluent was eluent A (5 m magnesium acetate, pH 9.0) and eluent B (methanol). The solvent gradient was set as follows: 2% B, 1.5 min; 2–100% B, 12.0 min; 100% B, 14.0 min; 100–2% B, 14.1 min; and 2% B, 17 min. The Q ExactiveTMHF-X mass spectrometer was operated in positive and negative polarity modes. The spray voltage was 3.2 kV, the capillary temperature was 320°C, the sheath gas flow rate was 40 arb, and the auxiliary gas flow rate was 10 arb. Compound Discoverer 3.1 (CD3.1, ThermoFisher) was used to process the raw data files generated by UHPLC-MS/MS to perform peak alignment, peak picking, and quantification for each metabolite.

### 4.3. Data Quality Control, Metabolite Pathway, and Classification Annotation

MetaX software (http://metax.genomics.cn, accessed on 23 July 2021) was used for the logarithmic conversion and standardization of the data [52]. The peaks extracted from all experimental samples and QC samples were analyzed by principal component analysis (PCA).

The main databases for the functional and classification annotation of the identified metabolites included the Kyoto Encyclopedia of Genes and Genomes (KEGG) (https://www.genome.jp/kegg/pathway.html, accessed on 23 July 2021), Human Metabolome Database (HMBD) (https://hmdb.ca/metabolites, accessed on 23 July 2021), and LIPID MAPS.

### 4.4. Screening of Differential Metabolites

The screening criteria for differential metabolites included the variable importance in the projection (VIP) of the first principal component of the partial least squares-discriminant analysis (PLS-DA) model VIP > 1.0 [53], and the *p*-value (significance at *p* < 0.05) was calculated by *t*-test [53,54,55,56]. Finally, the KEGG database was used to analyze the metabolic pathways of the differentially expressed metabolites (DEMs).

## 5. Conclusions

In order to investigate the metabolic changes of *H. ammodendron* and *H. persicum* under long-term drought conditions, non-targeted metabolomics analysis techniques were used to analyze their biological pathways and differential metabolites in response to drought. The results indicated that the differential metabolites in *H. ammodendron* and *H. persicum* are related to drought tolerance, but their regulation was different in the two species. Drought stress significantly promoted lipid degradation and metabolism in *H. ammodendron* and *H. persicum*, significantly increased the sugar metabolism, amino acid metabolism, biosynthesis and metabolism of nucleotide metabolites in *H. ammodendron*, as well as citric acid cycle, amino acid biosynthesis, and biosynthesis and metabolism of nucleotide metabolites in *H. persicum*. Both species can respond to long-term arid environments by regulating key metabolic pathways and metabolite synthesis. This study preliminarily revealed the molecular mechanism of the drought resistance of *H. ammodendron* and *H. persicum* at the metabolite level, and screened the key metabolites that respond to drought.

## Figures and Tables

**Figure 1 ijms-24-09099-f001:**
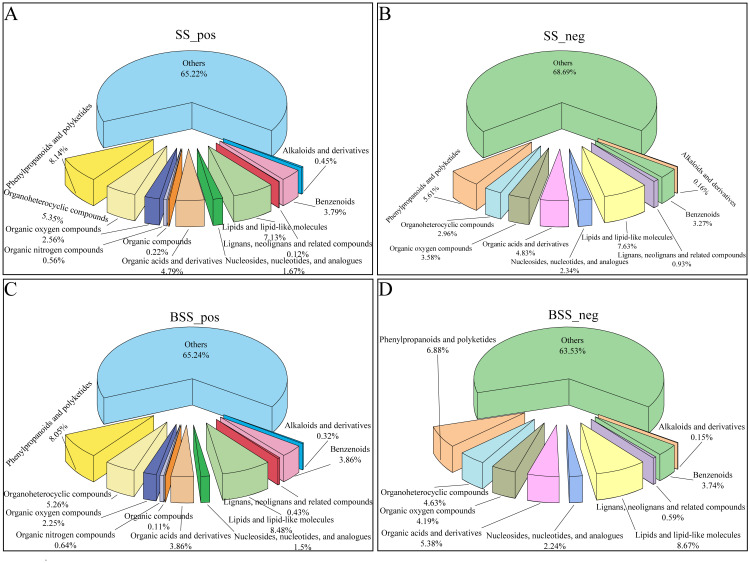
Distribution of different types of metabolites in *H. ammodendron* and *H. persicum*: (**A**) types of metabolites in *H. ammodendron* under positive ion mode (SS_pos); (**B**) types of metabolites in *H. ammodendron* under negative ion mode (SS_neg); (**C**) types of metabolites in *H. persicum* under positive ion mode (BSS_pos); and (**D**) types of metabolites in *H. persicum* under negative ion mode (BSS_neg).

**Figure 2 ijms-24-09099-f002:**
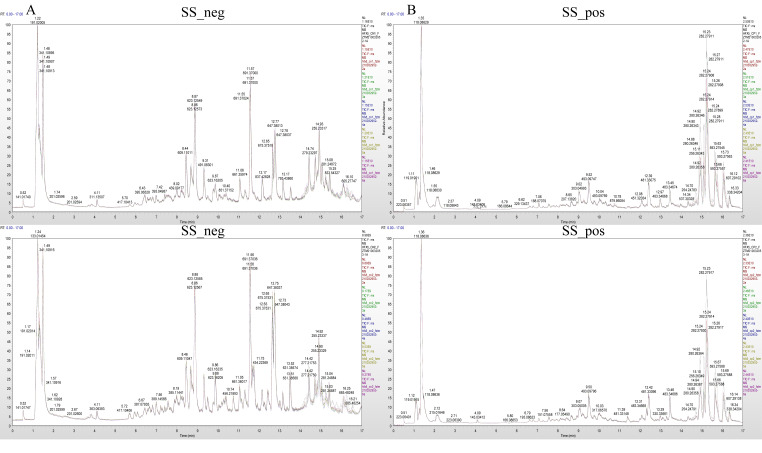
Chromatogram of total ion (TIC) of *H. ammodendron* in different soil moisture environments: (**A**) TIC diagram of *H. ammodendron* in negative ion mode (SS_neg); and (**B**) TIC diagram of *H. ammodendron* in positive ion mode (SS_pos).

**Figure 3 ijms-24-09099-f003:**
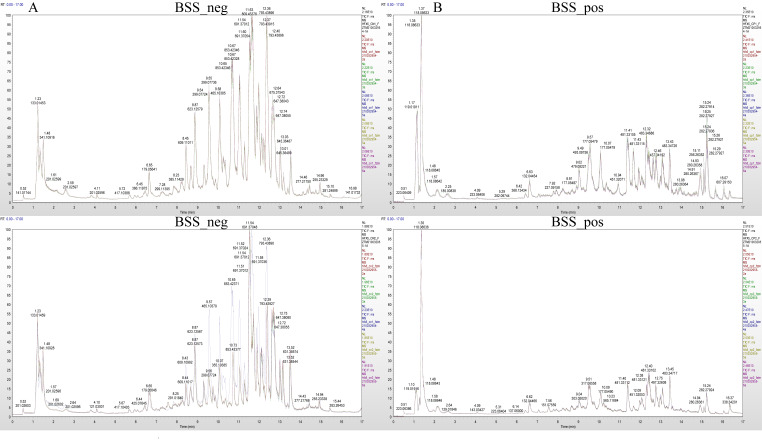
Chromatogram of total ion (TIC) of *H. persicum* in different soil moisture environments: (**A**) TIC diagram of *H. persicum* in negative ion mode (BSS_neg); and (**B**) TIC diagram of *H. persicum* in positive ion mode (BSS_pos).

**Figure 4 ijms-24-09099-f004:**
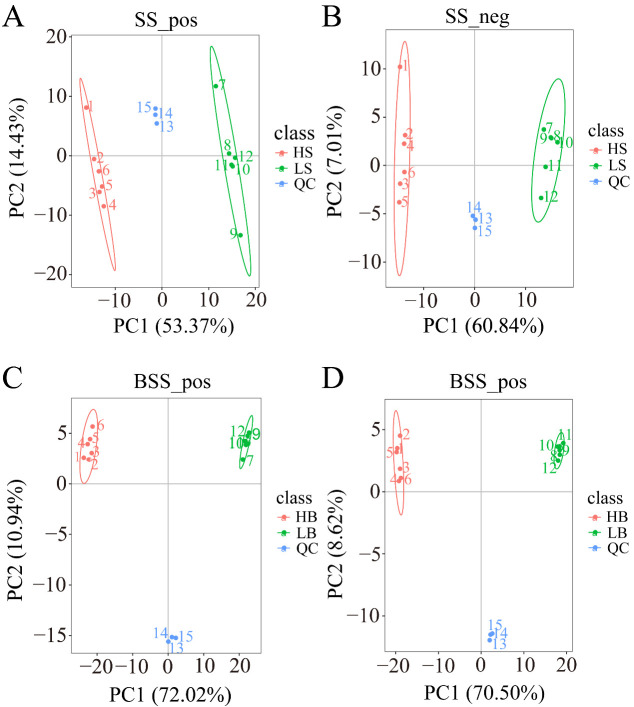
Principal component analysis (PCA) of the test samples and quality control samples: (**A**) PCA analysis of *H. ammodendron* samples and quality control samples in positive ion mode (SS_pos); (**B**) PCA analysis of *H. ammodendron* samples and quality control samples in negative ion mode (SS_neg); (**C**) PCA analysis of *H. persicum* samples and quality control samples in positive ion mode (BSS_pos); and (**D**) PCA analysis of *H. persicum* samples and quality control samples in positive ion mode (BSS_neg).

**Figure 5 ijms-24-09099-f005:**
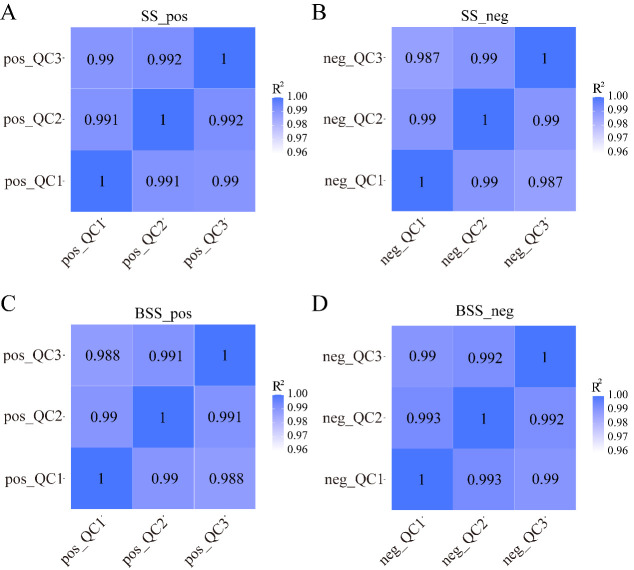
QC sample correlation analysis: (**A**) correlation analysis of *H. ammodendron* QC samples in positive ion mode (SS_pos); (**B**) correlation analysis of *H. ammodendron* QC samples in negative ion mode (SS_neg); (**C**) correlation analysis of *H. persicum* QC samples in positive ion mode (BSS_pos); and (**D**) correlation analysis of *H. persicum* QC samples in negative ion mode (BSS_neg). QC1, QC2, and QC3 represent three duplicate samples of quality control (QC) samples, respectively.

**Figure 6 ijms-24-09099-f006:**
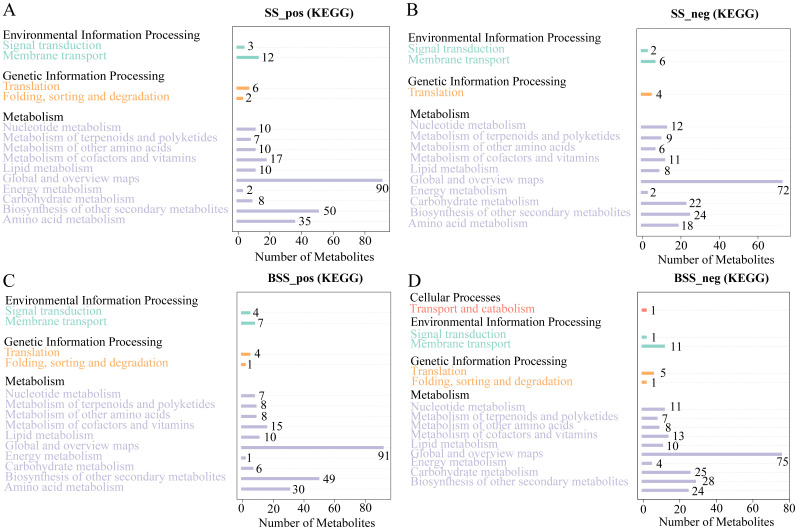
KEGG pathway annotation of all metabolites in *H. ammodendron* and *H. persicum*: (**A**) KEGG pathway annotation of all metabolites in *H. ammodendron* in positive ion mode (SS_pos); (**B**) KEGG pathway annotation of all metabolites in *H. ammodendron* in negative ion mode (SS_neg); (**C**) KEGG pathway annotation of all metabolites in *H. persicum* in positive ion mode (BSS_pos); and (**D**) KEGG pathway annotation of all metabolites in *H. persicum* in negative ion mode (BSS_neg).

**Figure 7 ijms-24-09099-f007:**
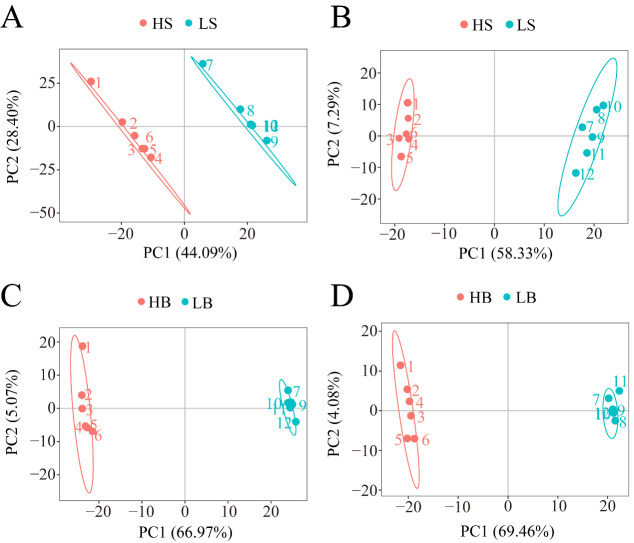
PCA analysis of the two comparative groups of *H. ammodendron* (LS vs. HS) and *H. persicum* (LB vs. HB) under drought stress: (**A**) PCA analysis of LS vs. HS in positive ion mode; (**B**) PCA analysis of LS vs. HS in negative ion mode; (**C**) PCA analysis of LB vs. HB in positive ion mode; and (**D**) PCA analysis of LB vs. HB in negative ion mode.

**Figure 8 ijms-24-09099-f008:**
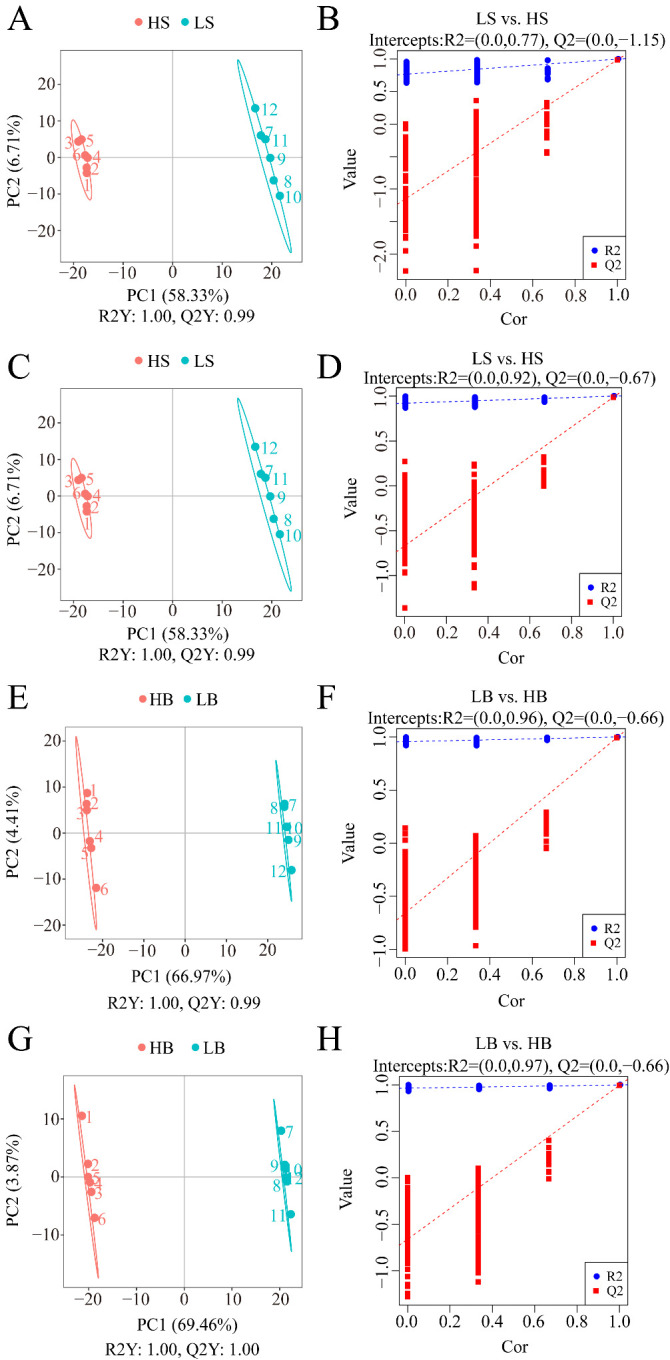
PLS-DA analysis of the two comparative groups of *H. ammodendron* and *H. persicum* under drought stress: (**A**) scatter plot of the PLS-DA scores of LS vs. HS in positive ion mode; (**B**) PLS-DA ranking verification diagram of LS vs. HS in positive ion mode; (**C**) scatter plot of PLS-DA scores of LS vs. HS in negative ion mode; (**D**) PLS-DA ranking verification diagram of LS vs. HS in negative ion mode; (**E**) scatter plot of the PLS-DA scores of LB vs. HB in positive ion mode; (**F**) PLS-DA ranking verification diagram of LB vs. HB in positive ion mode; (**G**) scatter plot of the PLS-DA scores of LB vs. HB in negative ion mode; and (**H**) PLS-DA ranking verification diagram of LB vs. HB in negative ion mode. Note: R2Y indicates the interpretation rate of the model, Q2Y is used to evaluate the prediction ability of the PLS-DA model, and an R2Y greater than Q2Y indicates that the model is well-established. The abscissa represents the correlation between Y of the random group and Y of the original group, and the ordinate represents the R2 and Q2 scores.

**Figure 9 ijms-24-09099-f009:**
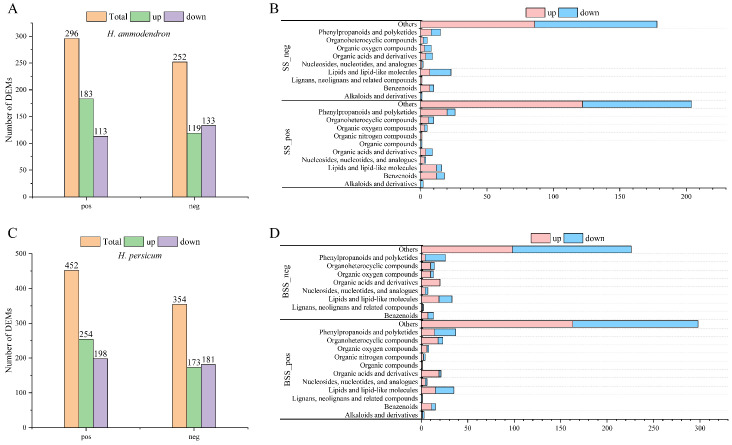
Statistics and material classification of the DEMs in LS vs. HS and LB vs. HB under drought stress: (**A**) statistics of the DEMs between LS vs. HS in positive ion (pos) and negative ion (neg) modes; (**B**) material classification of the DEMs between LS vs. HS in positive ion (SS_pos) and negative ion (SS_neg) modes; (**C**) statistics of the DEMs between LB vs. HB in positive ion (pos) and negative ion (neg) modes; and (**D**) material classification of the DEMs between LB vs. HB in positive ion (SS_pos) and negative ion (SS_neg) modes.

**Figure 10 ijms-24-09099-f010:**
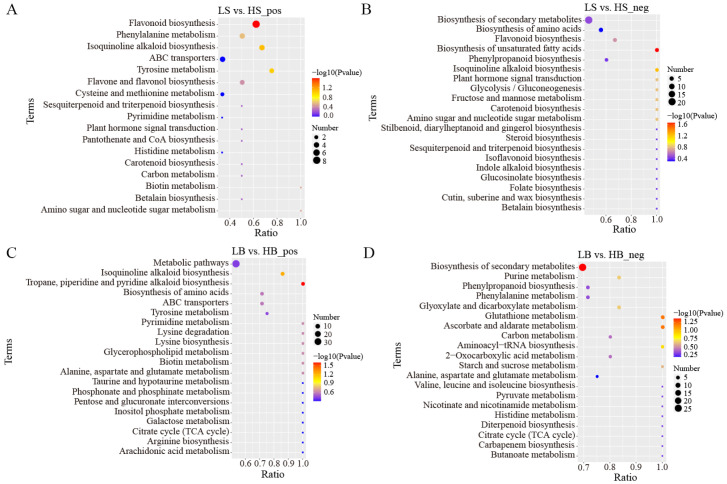
KEGG pathway enrichment analysis of DEMs in LS vs. HS and LB vs. HB under drought stress (only the top 20 results are shown): (**A**) KEGG pathway enrichment analysis of DEMs between LS vs. HS in positive ion mode; (**B**) KEGG pathway enrichment analysis of DEMs between LS vs. HS in negative ion mode; (**C**) KEGG pathway enrichment analysis of DEMs between LB vs. HB in positive ion mode; and (**D**) KEGG pathway enrichment analysis of DEMs between LB vs. HB in negative ion mode.

**Figure 11 ijms-24-09099-f011:**
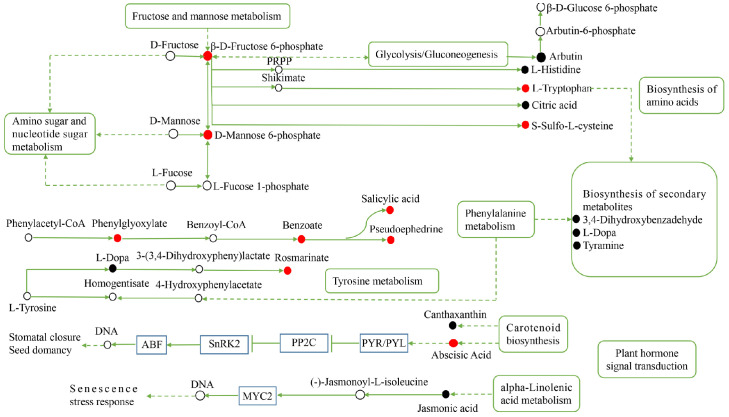
Construction of the key metabolic pathways of *H. ammodendron* in response to drought stress based on metabolomics analysis. Red circles represent the upregulated DEMs; black circles represent the downregulated DEMs and the white circles represent DEMs that are neither up nor down. The line ending with the bar indicates suppression; and the dashed arrows indicate indirect connections. PRPP: 5-phosphoribose-1-pyrophosphate.

**Figure 12 ijms-24-09099-f012:**
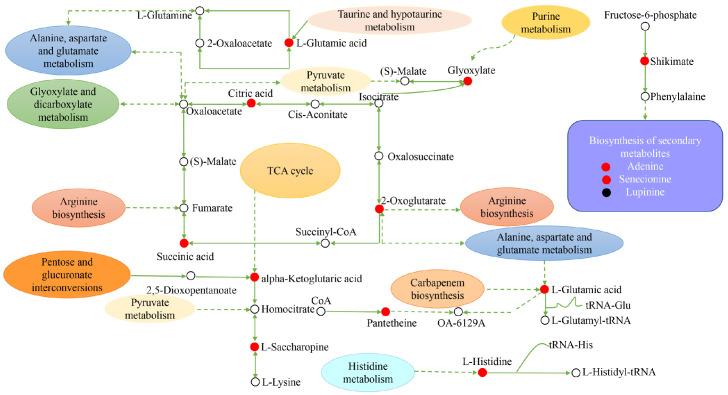
Construction of the key metabolic pathways of *H. persicum* in response to drought stress based on metabolomics analysis. Red circles represent the upregulated DEMs; black circles represent the downregulated DEMs and the white circles represent DEMs that are neither up nor down. Dashed arrows indicate indirect connections.

**Figure 13 ijms-24-09099-f013:**
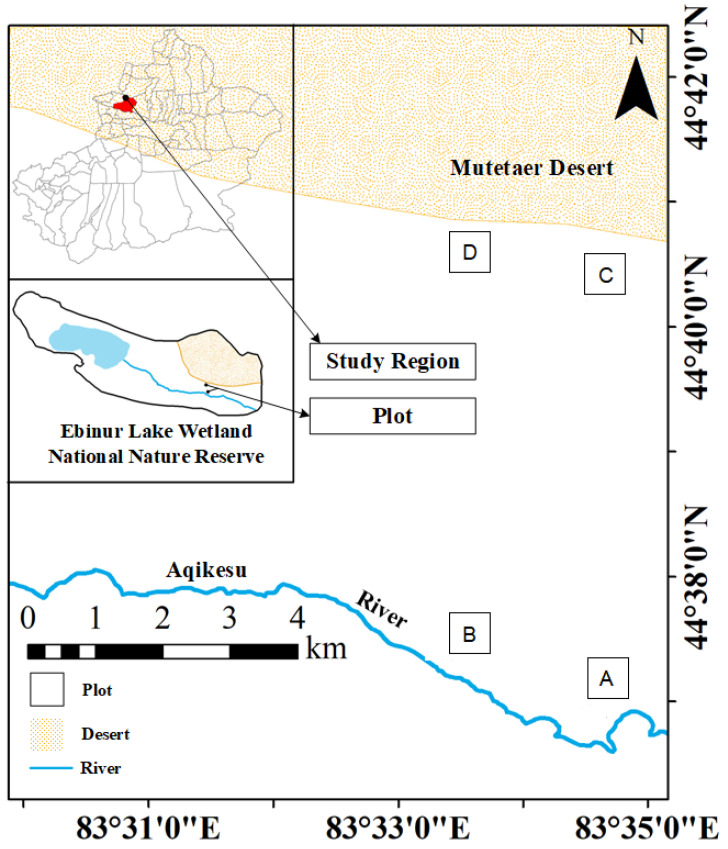
Study area and plot layout: the distribution area of *H. ammodendron*: A: humid and low salinity, and B: arid and low salt; and the distribution area of *H. persicum*: C: humid and low salinity, and D: arid and low salinity.

**Table 1 ijms-24-09099-t001:** KEGG annotation information of the DEMs in LS vs. HS and LB vs. HB.

Comparison Groups	DEMs	MapID
LS vs. HS_pos	L-Dopa	map00950, map00350, map00965
	Salicylic acid	map360, map04075
LS vs. HS_neg	β-D-Fructose 6-phosphate	map00010, map00051, map00520, map01110, map01230
	D-Mannose 6-phosphate	map00051, map00520, map01110
	Abscisic Acid	map00906, map04075, map01110
	L-Tryptophan	map01110, map00901, map00966, map01230
LB vs. HB_pos	alpha-Ketoglutaric acid	map00250, map00300, map00020, map00040, map00220, map00430
	L-Saccharopine	map00300, map01230, map00310, map01100
	L-Phenylalanine	map00960, map01230, map02010, map01100
LB vs. HB_neg	L-Glutamic acid	map01110, map00480, map00970, map00630, map00332, map00340, map00650, map00250
	Succinic acid	map00630, map00360, map00020, map00620, map00650, map00760, map00250
	Citric acid	map01110, map00230, map01200, map01210, map00020, map00250

## Data Availability

Data available for research upon request.

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
