# Peer review of "Metabolomic Analysis of the Response of Haloxylon ammodendron and Haloxylon persicum to Drought"

_ijms, 2023, doi:10.3390/ijms24109099_

Round 1

Reviewer 1 Report (Previous Reviewer 1)

I have no more comments and/or suggestions for authors.

All I had were presented and very well answered in the first submission of the manuscript.

Author Response

Thank you for taking the time to review this article again. Thank you very much for your previous suggestions, which greatly improved the quality of the article.
The revised article will be uploaded as an attachment for your review.

Reviewer 2 Report (New Reviewer)

The paper entitled “metabolomic analysis of the response of Haloxylon ammodendron and Haloxylon persicum to drought is a thorough study based on mass spectrometry attempting to characterize the metabolomic profiles of the two species under drought stress.

The work is experimentally and statistically well performed. In addition, Figures 12 and 13 show a clear comparative schematic representation of the conclusions facilitating the reader to understand the main points of the study.

There are a few points I would like to clarify related to the manuscript.

1. Discussion: Par. 362. “the content of organic…..”. Please clarify what is meant by “increase” i.e. what is compared with what and what symbols are used to indicate these conditions in the Results figures.

2. Please clarify what the QCs are and describe them

3. What is the difference between “arid and low salinity” and “drought and low salinity” ?

4. Please clarify figure 6. What are QC1, QC2 and QC3?

5. What is the difference between figure 5 and figure 8. Could the results of these two figures be shown in one?

6. Please explain what PLS-DA is.

7. Please clarify what are the numbers indicated inside the graphs of the PCA analysis.

8. Would it be useful a PCA comparison between the two species?

In general the paper describes a well conducted study on metabolic analysis during drought stress in two known stress resistant species.

I have no comments on the quality of the English language

Author Response

I would like to thank you for your careful and insightful comments on the manuscript. We have explained the opinions you mentioned and made modifications in the text. We hope that editors and reviewers can see our efforts.

Point 1:  Discussion: Par. 362. “the content of organic…..”. Please clarify what is meant by “increase” i.e. what is compared with what and what symbols are used to indicate these conditions in the Results figures.

Response 1:

We are grateful for reviewer’s guidance.

According to suggestions of reviewer, my explanation was as follows:

We classified the identified differential metabolites, where “up” represents upregulated differential metabolites, i.e., differential metabolites with increased content in the plant body; “down” indicates a downregulated differential metabolite, that is, a differential metabolite with reduced content in the plant body.

In Figures 10B and 10D, pink represents a class of upregulated differential metabolites, while blue represents a class of downregulated differential metabolites. For example, in the positive ion mode, all organic compound metabolites in H. persicum were upregulated, indicating an increase in the content of organic compound metabolites in H. persicum.

Point 2:  Please clarify what the QCs are and describe them.

Response 2:

We are grateful for reviewer’s guidance.

According to suggestions of reviewer, I would like to clarify the following:

Quality Control (QC) is a necessary step in obtaining stable and accurate metabolomic results. Especially when the sample size is large, it takes a certain amount of time for the sample to be tested on the machine, and the stability of the instrument and the normal signal response strength during the metabolite detection process are particularly important. Quality control can detect anomalies in a timely manner and solve problems as soon as possible to ensure the quality of the final collected data.

When performing standardization calibration, all samples to be collected can be mixed in equal amounts to obtain quality control (QC) samples.

Point 3:  What is the difference between “arid and low salinity” and “drought and low salinity”?.

Response 3:

We are grateful for reviewer’s guidance.

According to suggestions of reviewer, I want to express the same meaning. I should maintain consistency when writing. The writing of the article has been revised.

Point 4:  Please clarify figure 6. What are QC1, QC2 and QC3?

Response 4:

We are grateful for reviewer’s guidance.

According to suggestions of reviewer, I would like to clarify the following:

QC1, QC2, and QC3 represent three duplicate samples of quality control (QC) samples, respectively.

Point 5:  What is the difference between figure 5 and figure 8. Could the results of these two figures be shown in one?

Response 5:

We are grateful for reviewer’s guidance.

According to suggestions of reviewer, I would like to clarify the following:

Figure 5 shows the PCA analysis of all experimental and QC samples to reflect the stability of the method and the quality of the data. Figure 8 shows PCA analysis of two sets of experimental samples from two species before identifying differential metabolites, mainly to observe the overall distribution trend between the two sets of experimental samples. For subsequent identification and analysis of differential metabolites.

Point 6:  Please explain what PLS-DA is.

Response 6:

We are grateful for reviewer’s guidance.

According to suggestions of reviewer, I would like to clarify the following:

Partial Least Squares Discrimination Analysis (PLS-DA) is a supervised statistical method for discriminant analysis. This method uses partial least squares regression to establish a relationship model between metabolite expression levels and sample categories, in order to achieve prediction of sample categories. Establish PLS-DA models for each comparative group, and obtain model evaluation parameters (R2, Q2) through 7-fold cross validation (when the sample biological repetition number n<=3, k=2n). If R2 and Q2 are closer to 1, it indicates that the model is more stable and reliable.

Point 7:  Please clarify what are the numbers indicated inside the graphs of the PCA analysis.

Response 7:

We are grateful for reviewer’s guidance.

According to suggestions of reviewer, I would like to clarify the following:

1, 2, 3, 4, 5, and 6 replicate samples were metabolized in humid and low salt environments, while 7, 8, 9, 10, 11, and 12 represent 6 replicate samples in drought and low salt environments.

Point 8:  Would it be useful a PCA comparison between the two species?

Response 8:

We are grateful for reviewer’s guidance.

According to suggestions of reviewer, I would like to clarify the following:

We did not perform PCA analysis between two species, but instead conducted PCA analysis on samples from each species under two different soil moisture environments. For example, we conducted PCA analysis on the experimental samples of H. ammodendron under humid low salt and arid low salt conditions, and also conducted PCA analysis on the experimental samples of H. persicum under humid low salt and arid low salt conditions.

The revised article will be uploaded as an attachment for your review. Thank you again for your valuable feedback on this article. I will upload a revised version of the file for your review.

This manuscript is a resubmission of an earlier submission. The following is a list of the peer review reports and author responses from that submission.

Round 1

Reviewer 1 Report

The manuscript “Metabolomic analysis of the response of Haloxylon ammodendron and Haloxylon persicum to drought” provides very interesting results about the metabolic response of these drought tolerant desert plants to severe water deficit. 

Overall, the manuscript is very well written and presented, and implicate a detailed analysis of many results. However, I think the authors could improve the references, including bibliography more recent and with the species studied. I suggest authors could, at least, include evidence given in two recent manuscripts: an expert view about “The metabolic response to drought”, doi:10.1093/jxb/ery437, and the information about the main source of the water used by Haloxylon ammodendron and Haloxylon persicum, 10.3389/fpls.2022.804786

A more detailed revision was given directly in the manuscript reviewed uploaded.

Author Response

Dear reviewer,

We have made revisions to the proposed review comments one by one in the article, and made revisions to the content of this article based on the two articles, "The metabolic response to draft," doi: 10.1093/jxb/ery437, and the information about the main source of the water used by Haloxylon ammodendron and Haloxylon persistent, 10.3389/fpls.2022.804786.

I will upload the specific modifications in the form of a revised version for your review.

Thank you again for your valuable feedback on this article.

Reviewer 2 Report

The manuscript for “Metabonomic analysis of the response of Haloxylon ammodendron and Haloxylon persicum to drought” is well presented and provide in depth analysis of metabolomics in desert plants.

To improve the quality of the manuscript, I would suggest clarifying few points as bellow:

1)      In page 1, authors stating, “H. ammodendron exhibited 296 and 252 DEMs in positive and negative modes respectively, whereas 452 and 354 DEMs were identified in the positive and negative ion modes in H. persicum, respectively.” I would like to see the list of those metabolites, as supplementary data, so readers can learn the possible metabolite profile in this plant group. We can find only the list of few metabolite (around 40 to 60 metabolites), based on the fold changes significantly in the supplementary data sheet.

2)      Following the annotation of metabolites, some of the metabolites stated in the supplementary data file has given either “no results” or “invalid mass) and still those metabolites has taken to account. As an example, “Amoxicillin” is given in the list and can you clarify the existence of those metabolites in plants?

3)      In Figure 5, PCA of the samples and QC are showing nice correlation between HS, LS and QC (A and B), what is the reason we don’t see QC are not clustering in between HB and LB? Any reason to get QCs are prepared not from the HB and LB?  It would be great to get some explanation of the QC behaviour in C and D.

4)      Regarding the lipid metabolites (Page 15), does the metabolite extraction protocol is sufficient to extract lipids? Authors discussed only six glycerol phospholipids metabolites are significant, but this may be due the extraction protocol you have used, which is not strong enough to extract all the lipid species, so you will detect few fatty acids and lyso glycerol phospholipids. Please revised the 3.2 section as the current statement is misleading to readers.

Author Response

I would like to thank you for your careful and insightful comments on the manuscript. Based on your comments, we have made large-scale revisions and rewriting of the manuscript. We hope that editors and reviewers can see our efforts.

Point 1:  In page 1, authors stating, “H. ammodendron exhibited 296 and 252 DEMs in positive and negative modes respectively, whereas 452 and 354 DEMs were identified in the positive and negative ion modes in H. persicum, respectively.” I would like to see the list of those metabolites, as supplementary data, so readers can learn the possible metabolite profile in this plant group. We can find only the list of few metabolite (around 40 to 60 metabolites), based on the fold changes significantly in the supplementary data sheet.

Response 1:

We are grateful for reviewer’s guidance.

According to suggestions of reviewer, we uploaded the differential metabolites of alfalfa and persimmon identified by positive and negative ion modes as attachments (Table S2 and Table S3), and provided annotations in the article. (Page 10)

Point 2: Following the annotation of metabolites, some of the metabolites stated in the supplementary data file has given either “no results” or “invalid mass) and still those metabolites has taken to account. As an example, “Amoxicillin” is given in the list and can you clarify the existence of those metabolites in plants?

Response 2:

We are grateful for reviewer’s guidance.

According to suggestions of reviewer, I would like to clarify the following:

In addition to annotating the identified differential metabolites into universal metabolomics databases (such as KEGG, HMDB, etc.), we also annotate them into mzCloud, mzVault, and MassList databases. These three types belong to network databases, and there may be situations where some metabolites can be annotated while others cannot.

Point 3: In Figure 5, PCA of the samples and QC are showing nice correlation between HS, LS and QC (A and B), what is the reason we don’t see QC are not clustering in between HB and LB? Any reason to get QCs are prepared not from the HB and LB?  It would be great to get some explanation of the QC behaviour in C and D.

Response 3:

We are grateful for reviewer’s guidance.

According to suggestions of reviewer, I would like to clarify the following:

Quality Control (QC): In order to obtain reliable and high-quality metabolomic data in mass spectrometry based metabolomics research, quality control (QC) is usually required. QC is only a display of sample test results, and does not screen or "flush" data.

Therefore, QC identifies problems and checks for any abnormalities during the sample testing process. The more QC is summarized, the more stable the method is, and the higher the data quality is.

Point 4: Regarding the lipid metabolites (Page 15), does the metabolite extraction protocol is sufficient to extract lipids? Authors discussed only six glycerol phospholipids metabolites are significant, but this may be due the extraction protocol you have used, which is not strong enough to extract all the lipid species, so you will detect few fatty acids and lyso glycerol phospholipids. Please revised the 3.2 section as the current statement is misleading to readers.

Response 4:

We are grateful for reviewer’s guidance.

According to suggestions of reviewer, we made modifications to section 3.2 and checked for language expression issues.

Thank you again for your valuable feedback on this article. I will upload a revised version of the file for your review.

Round 2

Reviewer 1 Report

Thank you for the improvement.

The manuscript can be accept

Best regards